# Diversity Matters: User-Centric Multi-Interest Learning for Conversational Movie Recommendation

## ABSTRACT

Diversity plays a crucial role in Recommender Systems (RSs) as it ensures a wide range of recommended items, providing users with access to new and varied options. Without diversity, users often encounter repetitive content, limiting their exposure to novel choices. While significant efforts have been dedicated to enhancing recommendation diversification in static offline scenarios, relatively less attention has been given to online Conversational Recommender Systems (CRSs). However, the lack of recommendation diversity in CRSs will increasingly exacerbate over time due to the dynamic user-system feedback loop, resulting in challenges such as the Matthew effect, filter bubbles, and echo chambers. To address these issues, we propose an innovative end-to-end CRS paradigm called User-Centric Multi-Interest Learning for Conversational Movie Recommendation (CoMoRec), which aims to learn user interests from multiple perspectives to enhance result diversity as users engage in natural language conversations for movie recommendations. Firstly, CoMoRec automatically models various facets of user interests, including context-based, graph-based, and review-based interests, to explore a wide range of user intentions and preferences. Then, it leverages these multi-aspect user interests to accurately predict personalized and diverse movie recommendations and generate fluent and informative responses during conversations. Through extensive experiments conducted on two publicly available CRS-based movie datasets, our proposed CoMoRec achieves a new state-of-the-art performance and outperforms all the compared baselines in terms of improving recommendation diversity in the CRS.

## CCS CONCEPTS

• **Information systems** → **Recommender systems**.

## KEYWORDS

Conversational Recommender System, Diversified Movie Recommendation, Multi-Interest User Preferences, Natural Language Conversations

**ACM Reference Format:**
Anonymous Author(s). 2024. Diversity Matters: User-Centric Multi-Interest Learning for Conversational Movie Recommendation. In *Proceedings of Make sure to enter the correct conference title from your rights confirmation emai (Conference acronym 'XX)*. ACM, New York, NY, USA, 9 pages. https://doi.org/XXXXXXX.XXXXXXX

## 1 INTRODUCTION

With the rapid development of intelligent agents, Conversational Recommender Systems (CRSs) [16, 17, 21, 23, 35, 45, 46] has emerged as a prominent research topic, aiming to provide effective recommendations by engaging in natural language conversations between the user and the system. CRSs have been broadly adopted in various domains like music recommendation [10], electric commerce [20], and health counseling [31], *etc*. Despite these advancements, CRSs still face the challenge of inadequate recommendation diversity, and this problem becomes more pronounced as user interactions prolong. The lack of diverse recommendations can give rise to significant issues like exposure bias, filter bubbles, and echo chambers. Thus, enhancing diversification is crucial for the success of CRSs.

Recently, many research efforts have been devoted to facilitate the diversified recommendation. These endeavors can be classified into three primary directions: 1) Post-Processing (PP) methods [1, 3–5, 25, 47] involve the addition of a re-ranking or post-processing module to recommended items to strike a balance between relevance and diversity. 2) Determinantal Point Process (DPP) methods [6, 11, 13, 14, 37, 38] aim to select a diverse subset of items from a larger pool of retrieved items, utilizing heuristics different from those used in PP-based methods. 3) Learning To Rank (LTR) methods [8, 18] enhance recommendation diversity by optimizing the ranking strategy to generate an ordered list of items instead of a candidate set. While these methods have made significant strides in improving recommendation diversity, both PP-based methods and DDP-based ones rely heavily on the quality of user and item representations, whereas LTR-based methods face challenges in acquiring appropriate datasets. To this end, numerous graph-based algorithms [19, 32, 33, 39] have emerged to expand the spectrum of diverse items. By constructing the user-item bipartite graphs, these algorithms facilitate greater access to a wide array of diverse items.

Despite their effectiveness, most existing methods still encounter two major issues: *1) Diversity Exploration.* While many current methods [1, 6, 8, 19] primarily focus on exploring recommendation diversity in offline settings with relatively static conditions, there exists a noticeable research gap when it comes to enhancing result diversity in interactive CRS contexts. In practical scenarios, insufficient recommendation diversity can lead to users being exposed to repetitive content, limiting their access to novel and varied options. Moreover, the issue of recommendation diversity becomes more prominent as users interact with the system over time [41], giving rise to notorious problems such as exposure bias [36], filter bubbles [27], and echo chambers [12]. Therefore, improving recommendation diversification in CRSs is of utmost importance. *2) Graph Structure.* Many graph-based algorithms [19, 32, 33, 39] strive to enhance the coverage of diverse items by constructing user-item bipartite graphs. However, these graphs often encounter sparsity issues, as a significant proportion of users engage with or express preferences for only a limited subset of items. This sparse graph structure presents challenges in capturing meaningful relationships

or patterns between users and items. Moreover, bipartite graphs are not ideally suited for capturing higher-order relationships or interactions that involve more than two sets of entities. Representing complex relationships that extend beyond pairwise interactions becomes challenging within the confines of bipartite graphs. Consequently, this limitation can impact the quality of recommendations and the overall user experience.

To address these issues, we propose a novel end-to-end paradigm, User-Centric Multi-Interest Learning for **Co**nversational **Mo**vie **Rec**ommendation (**CoMoRec**), which is comprised of User-Centric Multi-Interest Learning and Interest-Enhanced CRS, aiming to modeling multi-aspect user interests for improving recommendation diversity as users interact with the system over time in the CRS. Considering the sparsity issue of the traditional user-item bipartite graphs, User-Centric Multi-Interest Learning paradigm first devises a high-order and densely connected Temporal Knowledge Graph (TKG) by leveraging the large-scale DBpedia Knowledge Graph (KG) as a valuable repository of structured entity data. Specifically, we extract entities from the conversation context and item reviews as the seed set at each time step. Based on these seed entities, we traverse the DBpedia KG to collect one-hop triples that where these triples consist of head-relation-tail associations that provide meaningful connections between the entities. After constructing the TKG, User-Centric Multi-Interest Learning paradigm further models multi-aspect user interests including context-based, graph-based, and review-based interest to capture the wide array of user intentions and preferences by adopting historical conversations, TKG, and item reviews. Moreover, the Interest-Enhanced CRS module focuses on leveraging these multiple user interests to make conversational movie recommendation. Concretely, it excels in making precise predictions for personalized and diverse movie recommendations that align with users' intentions and interests in the recommendation task, and generating fluent and informative responses in the conversation task. By leveraging the information obtained from context-based, graph-based, and review-based interests, the system can produce more tailored and engaging dialogue, fostering a positive user experience. Extensive experiments conducted on two publicly CRS-based movie datasets have provided compelling evidence of the superiority of our proposed CoMoRec for conversational movie recommendation, and the effectiveness of improving recommendation diversity in the CRS.

Overall, our main contributions are included as follows:

- To the best of our knowledge, this is the first work to model multi-aspect user interests, including context-based interest, graph-based interest, and review-based interest, to improve recommendation diversity as the user continually interacts with the system over time in the CRS.
- We propose a novel end-to-end paradigm, CoMoRec, which is comprised of User-Centric Multi-Interest Learning and Interest-Enhanced CRS. The former aims to automatically model multi-aspect user interests while the latter devotes to accurately predict items and effectively generate responses.
- Extensive experiments on two CRS-based movie datasets show the superiority of our CoMoRec, which demonstrates the effectiveness of our proposed method in improving recommendation diversification in the CRS.

## 2 RELATED WORK

### 2.1 Conversational Recommender System

With the rapid development of intelligent agents in various domains, Conversational Recommender Systems (CRSs) have attracted a lot of attention from researchers [16, 17, 21, 23, 35, 45, 46], which aim to provide accurate recommendation through natural language conversations between users and systems [21, 46]. These CRS-based methods can be divided into two groups: attributed-based CRS and human-like CRS. The former [9, 28, 42, 44] aims to ask users which attributes they like or dislike for efficiently explore user preference by leveraging pre-defined actions (*e.g.*, item attributes and intent slots). These methods usually utilize the multi-armed bandit models [9] or reinforcement learning [28] to optimize the interaction strategy. Due to the heavy dependencies on the pre-defined actions and templates, they cannot be flexibly applied in various domains. The latter [16, 17, 21, 45, 46] is more realistic because they tend to provide recommendation via human-like responses. Human-like CRS usually designs a conversation module to provide proper response and a recommendation module to make recommendations. However, these approaches suffer from the limited and inadequate contextual information in the initial conversational utterances. To address these problems, most existing methods either introduce structured external data (*e.g.*, knowledge graph) [43, 46], or unstructured external data (*e.g.* item reviews) [21], to complement the conversation utterance. These approaches still fall short in terms of providing diversified recommendations. Instead, we follow the latter category and model multi-aspect user interests by leveraging various knowledge to improve recommendation diversity as user interacts with the system over time in the CRS.

### 2.2 Diversified Recommendation

The concept of recommendation diversification was initially introduced by Ziegler et al. [47], who employed a greedy algorithm [5] inspired by the field of information retrieval. Since then, a series of Post-Processing (PP)-based methods [1, 3–5, 25, 47] have been proposed to achieve a balance between recommendation relevance and diversity. For instance, Sha et al. [25] propose an advanced framework that incorporates the notions of relevance, user preferences, and variety. Similarly, Qin et al. [22] address this issue by employing a linear combination of the rating function and an entropy regularizer. Later, Determinantal Point Processes (DPP)-based methods [6, 11, 13, 14, 37, 38] focus on selecting a diverse subset of items from a larger pool of retrieved items, replacing the heuristics employed in PP-based methods. For instance, Gartrell et al. [11] introduce a novel approach to learning the DPP kernel from observed data by employing a low-rank factorization of the kernel. Recently, a new line of methods based on Learning to Rank (LTR) [8, 18] has emerged, aiming to enhance recommendation diversity through the adoption of ranking strategies. For instance, Cheng et al. [8] put forward a machine learning-based diversification approach by integrating the recommendation model with a structured Support Vector Machine (SVM) [29]. Despite their effectiveness, these existing methods focus on the recommendation diversity in the offline recommendation settings, instead, our proposed work aims to enhance recommendation diversification as users chat with the system over time in the CRS.

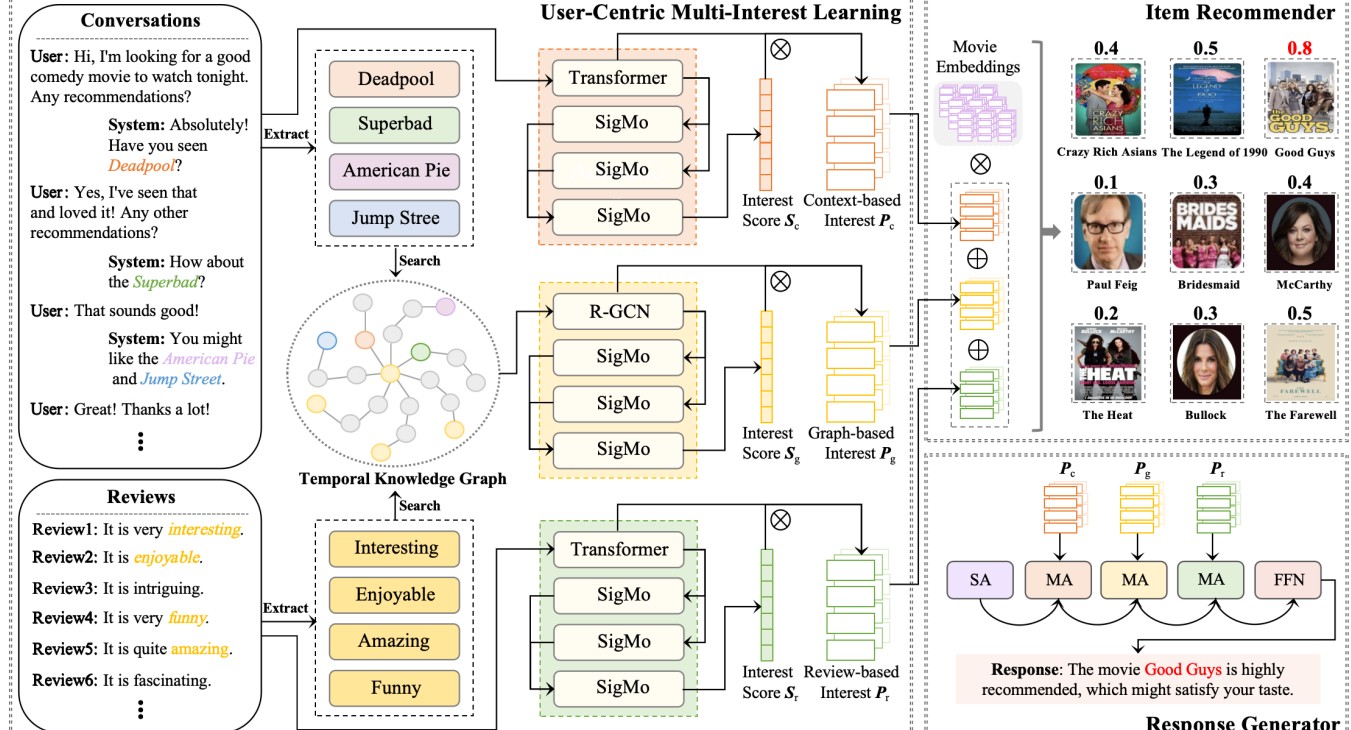

**Figure 1: Overview of the proposed framework, CoMoRec, which is comprised of User-Centric Multi-Interest Learning and Interest-Enhanced CRS. The former devotes to adaptively model multi-aspect user interests including context-based, graph-based, and review-based interests by incorporating conversations, temporal knowledge graph, and item reviews; the latter aims to make diverse movie predictions in the recommendation task (*i.e.*, item recommender) and generate informative responses in the conversation task (*i.e.*, response generator) by employing these learned multi-aspect user interests.**

## 3 COMOREC

In the CRS, as users continually with the online system over time, if the lack of recommendation diversification persists, it can lead to a series of notorious issues such as filter bubbles and echo chambers. To address these issues, we propose a novel paradigm CoMoRec, which is comprised of User-Centric Multi-Interest Learning and Interest-Enhanced CRS. The former focuses on modeling multi-aspect user interests, while the latter aims to adopt these multiple interests to accurately predict items in the recommendation task and effectively generate responses in the conversation task. The pipeline of our CoMoRec is depicted in Fig.1.

### 3.1 User-Centric Multi-Interest Learning

*3.1.1* **Temporal Knowledge Graph**. Most conventional methods strive to explore the diverse range of user interests by constructing user-item bipartite graphs, which establish connections between users and items based on their historical interaction logs. However, these bipartite graphs often face the challenge of sparsity since the majority of users tend to interact with or express preferences for only a small subset of items. Therefore, capturing meaningful relationships or patterns between users and items becomes difficult in the face of such sparsity. To address these issues, we propose the

Temporal Knowledge Graph (TKG) to build higher-order structural connectivity by leveraging large-scale knowledge graph.

**Context-based Entities Extraction.** To construct the TKG, we first extract entities from the conversations by retrieving the entity names over the large-scale DBpedia [2] KG $\mathcal{G}$ due to its fruitful facts and relations. It consists of a large number of triples $(e_1, r, e_2)$, where $e_1$ and $e_2 \in \mathcal{E}$ refer to the head and tail entities, and $r$ denotes the relation between them. Let $C = \{s_t\}_{t=1}^n$ denote the conversation context, comprising all utterances $s_t$ that form the dialogue history provided by the user and the system in alternating turns. Firstly, we establish a mapping between each item in the item set $\mathcal{V}$ and the corresponding entity in the entity set $\mathcal{E}$ using their names, inspired by [43]. For example, the movie item "The Heat" mentioned in the $C$ would be linked to "http://dbpedia.org/resource/The_Heat_(film)" in the DBpedia KG $\mathcal{G}$. Besides, we utilize a similar approach to associate informative non-item entities that appear in $C$ with entities within $\mathcal{E}$. This step assists in identifying relevant entities that are connected to the items and conversation responses. Moreover, we perform entity linking on the conversation history, which involves identifying and extracting entities mentioned within the conversation. Formally, at each time step $t$, context-based entity set $\mathcal{E}_c^{(t)}$ can be described as:

$$\mathcal{E}_c^{(t)} = \mathcal{F}_{\text{extract}}(C, \mathcal{V}, \mathcal{E}, \mathcal{G}). \tag{1}$$

**Review-based Entities Extraction.** Next, our attention turns to the crucial task of extracting entities from the relevant reviews. It is important to note that our primary objective is to identify and retrieve reviews that provide valuable insights, as not all reviews contain meaningful information. In fact, irrelevant reviews can hinder the exploration of diverse user interests. Additionally, reviews expressing inconsistent attitudes can introduce noise into the discussion, making it challenging to generate coherent responses. To this end, we aim to source important and useful reviews that are coherent with the ongoing conversation [21].

Concretely, when considering the entire set of reviews $\mathcal{R}$, the key is to select those reviews that exhibit a similar sentiment polarity to the conversational history. Suppose that $\mathcal{R}_v^{(c)} = \{r_1, r_2, \cdots, r_n\} \in \mathcal{R}$ represents the reviews associated with the item mentioned in the conversation history $C$. For each review $r_j = \{w_1, w_2, \cdots, w_m\} \in \mathcal{R}_v^{(c)}$, we employ a transformer-based sentiment predictor [21] to predict its sentiment polarity. Here, $\mathcal{H}^{(l-1)}(r_j)$ represents the output embeddings of the previous transformer layer, and the output of the current layer $\mathcal{H}^{(l)}(r_j)$ can be defined using the *Multi-head Attention Function* MHA$(\cdot, \cdot, \cdot)$ as follows:

$$\mathcal{H}^{(l)}(r_j) = \text{MHA}(\mathcal{H}^{(l-1)}(r_j), \mathcal{H}^{(l-1)}(r_j), \mathcal{H}^{(l-1)}(r_j),$$
$$\text{MHA}(K, Q, V) = [h_1^l, h_2^l, \cdots, h_h^l]W_j^l,$$
$$h_j^l = \text{SA}(\mathcal{H}^{(l)}(r_j)W_j^k, \mathcal{H}^{(l)}(r_j)W_j^q, \mathcal{H}^{(l)}(r_j)W_j^v), \quad (2)$$
$$\text{SA}(K, Q, V) = \text{Softmax}(\frac{QK^T}{\sqrt{d/h}})V.$$

Here $r_j = \{w_1, w_2, \cdots, w_m\}$ is the embedding of the review $r_j$, and $w_i$ denotes the embedding of each word $w$, $h$ represents the number of heads, $W_j^l$ is a learned parameter during model training, and each head $h_j^l$ is calculated using the attention mechanism SA$(\cdot, \cdot, \cdot)$. In this attention mechanism, $K$, $Q$, and $V$ denote the key, query, and value matrices, respectively, while $W_j^k$, $W_j^q$, and $W_j^v$ are trainable parameters. For simplicity, we consider the output embeddings of the top transformer layer as the final review representations $H$. Formally, this process can be described as follows:

$$H = \text{MHA}(\mathcal{H}^{(L-1)}(r_j), \mathcal{H}^{(L-1)}(r_j), \mathcal{H}^{(L-1)}(r_j),$$
$$P_v = \text{Softmax}(W_1 \tanh(W_2 H^T)). \quad (3)$$

$L$ represents the number of transformer layers, and $P_v$ denotes the predicted sentiment towards the movie $v$ in review $r_j$. Similarly, we use this transformer-based sentiment prediction to evaluate the sentiment polarity $P_v^*$ for the conversation sentence that mentions the movie $v$. Ultimately, we select reviews that share a similar sentiment polarity with the conversation sentence to establish the retrieved reviews $\widetilde{\mathcal{R}}_v^{(c)}$, which can be written as:

$$\widetilde{\mathcal{R}}_v^{(c)} = \{\tilde{r}_1, \tilde{r}_2, \cdots, \tilde{r}_{\tilde{n}}\},$$
$$\widetilde{\mathcal{R}}_v^{(c)} \in R_v^{(c)}, \tilde{n} << n. \quad (4)$$

Note that there are multiple reviews related to the item $v$ that are currently being discussed in the conversation history $C$. To simplify the process, we choose to select one sentence for each mentioned item. To achieve this, we employ a word-wise method to randomly select a set of words or phrases to construct each "sentence". While this strategy may affect the fluency of the sentences, it brings forth a substantial level of diversity given the extensive pool of words and phrases accessible. The definitive review, represented as $r_v$ (*i.e.*, one sentence), for each item $v$ mentioned in the conversation responses $C$ can be written as follows:

$$r_v = \mathcal{F}_{\text{retrieve}}(\widetilde{\mathcal{R}}_v^{(c)}, \mathcal{V}, \mathcal{W}, C), \quad (5)$$

where $\mathcal{F}_{\text{retrieve}}(\cdot)$ signifies the review retrieval function, $\mathcal{W}$ represents all the words in $\widetilde{\mathcal{R}}_v^{(c)}$. Upon acquiring the found review, $r_v$, it is incorporated into the review set $\mathcal{R}$, which houses reviews of items that have appeared in conversation $C$. Once we've garnered the pivotal reviews, we go on to extract the entities $\mathcal{E}_r$ from the selected set of reviews, $\mathcal{R}$. At the time step $t$, this can be articulated:

$$\mathcal{E}_r^{(t)} = \mathcal{F}_{\text{extract}}(\mathcal{R}, \mathcal{V}, \mathcal{E}, \mathcal{G}). \quad (6)$$

**Graph Relations Extraction.** Finally, we combine the conversation entities $\mathcal{E}_c^{(t)}$ and the review entities $\mathcal{E}_r^{(t)}$ by concatenating them, allowing us to model diverse user-item interactions and accurately capture structural information. This process can be represented as:

$$\mathcal{E}_g^{(t)} = (\mathcal{E}_c^{(t)} \oplus \mathcal{E}_r^{(t)}), \quad (7)$$

where $\oplus$ represents the concatenation operation. The resulting set of entities $\mathcal{E}_g^{(t)}$ is regarded as the seed set. Subsequently, we extract one-hop triples from the DBpedia graph using these seed entities, thereby constructing our target TKG $\mathcal{G}_t$, as shown below:

$$\mathcal{G}_t = \text{One-hop}(\mathcal{E}_g^{(t)}) = \{(e, r, e') \mid \text{Triple}(e, r, e') \in \mathcal{G}, \ e \in \mathcal{E}_g^{(t)}\}. \quad (8)$$

Here $e$ and $e'$ are the head and tail entities while $r$ is the relation between them.

### 3.1.2 *Multi-Interest Modeling*. 
After building the TKG, we model multi-aspect user interests (i.e., context-based interest, graph-based interest, and review-based interest) by utilizing conversation contexts, TKG, and item reviews, aiming to enhance recommendation diversification as users progressively interact with the system over time in the CRS.

**Context-based Interest.** In contrast to fixed user profiles or direct input from users, conversations offer a more detailed perspective on user preferences by capturing the ever-changing nature of their interests. The continuous exchange of dialogues presents an opportunity to comprehend the evolving interests of users. Taking into account the conversational context, sentiment, and the variety of discussed topics, the CRS can dynamically adjust its recommendations to cater to the user's current interests. This approach enables a more personalized and customized recommendation experience that aligns with the user's current preferences. To do this, we employ the Transformer as the encoder to efficiently encode the conversations and derive their corresponding representations inspired by the valuable attributes of the Transformer model [30]. Given a conversation context $C$, let $\mathcal{H}^{(l-1)}(C)$ be the output embeddings of the previous transformer layer, and the output of the current layer $\mathcal{H}^{(l)}(C)$ can be defined by the MHA$(\cdot, \cdot, \cdot)$ as:

$$\mathcal{H}^{(l)}(C) = \text{MHA}(\mathcal{H}^{(l-1)}(C), \mathcal{H}^{(l-1)}(C), \mathcal{H}^{(l-1)}(C)). \quad (9)$$

The specifics of the function MHA$(\cdot, \cdot, \cdot)$ can be found in Eq. (2). To streamline the procedure, we opt for the output embedding of the

final transformer layer as the ultimate representation, labeled as $\boldsymbol{H}_c$, and can be formally defined as:

$$\boldsymbol{H}_c = \text{MHA}(\mathcal{H}^{(\mathcal{L}_c-1)}(C), \mathcal{H}^{(\mathcal{L}_c-1)}(C), \mathcal{H}^{(\mathcal{L}_c-1)}(C)). \quad (10)$$

Here $\mathcal{L}_c$ represents the maximum number of transformer layers. Once we acquire the ultimate output representations $\mathcal{L}_c$, we employ them in three non-linear functions as outlined below to generate the context-based interest score:

$$\boldsymbol{S}_c = \text{Softmax}(\text{S}_{\text{op}}(\frac{\text{SigMo}[(\text{SigMo}(\mathbf{W}_q^{(c)}\boldsymbol{H}_c))(\text{SigMo}(\mathbf{W}_k^{(c)}\boldsymbol{H}_c))^T]}{\sqrt{d}})), \quad (11)$$

where the function Softmax represents the softmax function used to normalize the embeddings. The function $\text{S}_{\text{op}}(\cdot)$ involves the process of summing each row and subsequently averaging the rows within the embedding matrix, while masking the diagonal elements. $\text{SigMo}(\cdot)$ denotes the sigmoid function, and $\mathbf{W}_q^{(c)}$ and $\mathbf{W}_k^{(c)}$ are trainable parameters. Finally, we perform the interactions between the score $\boldsymbol{S}_c$ and $\boldsymbol{H}_c$ for generating the context-based interest as follows:

$$\boldsymbol{P}_c = \boldsymbol{S}_c \otimes \boldsymbol{H}_c. \quad (12)$$

**Graph-based Interest.** Due to the sparsity of the user-item bipartite graph, we merge conversations and reviews to create the TKG by integrating the comprehensive DBpedia KG. This enables us to harness the wealth of information present in both the conversations/reviews and the DBpedia KG. As relations within the TKG play a pivotal role in uncovering users' interests, we employ Relational Graph Convolutional Networks (R-GCNs) [21] to encode structural and relational details from the extracted subgraph $\mathcal{G}_t$ into entity representations. Formally, let $e$ represent a node at the $(l+1)$-th layer in $\mathcal{G}_t$, and its feature representation can be calculated as follows:

$$\boldsymbol{h}_e^{l+1} = \sigma\left(\sum_{r \in \mathcal{R}} \sum_{\hat{e} \in \mathcal{N}_e^r} \frac{1}{z_{e,r}} \boldsymbol{W}_r^l \boldsymbol{h}_{\hat{e}}^l + \boldsymbol{W}^l \boldsymbol{h}_e^l\right), \quad (13)$$

Here $\boldsymbol{h}_e^l \in \mathbb{R}^d$ represents the hidden representation of entity $e$ at the $l$-th layer of the graph neural network, where $d$ denotes the feature dimensionality. The set $\mathcal{N}_e^r$ refers to the one-hop neighbor set of entity $e$ under the relation $r$. The hyperparameter $z_{e,r}$ corresponds to the normalization factor. The matrices $\boldsymbol{W}_r^l$ and $\boldsymbol{W}^l$ are trainable parameters that are updated during the model training process. These matrices are used to transform and update the hidden representations of entities within the graph neural network. Let $\boldsymbol{H}_g$ be the final output embedding of R-GCN, then we can obtain the graph-based interest score:

$$\boldsymbol{S}_g = \text{Softmax}(\text{S}_{\text{op}}(\frac{\text{SigMo}[(\text{SigMo}(\mathbf{W}_q^{(g)}\boldsymbol{H}_g))(\text{SigMo}(\mathbf{W}_k^{(g)}\boldsymbol{H}_g))^T]}{\sqrt{d}})). \quad (14)$$

Similarly, we can induce the graph-based interest:

$$\boldsymbol{P}_g = \boldsymbol{S}_g \otimes \boldsymbol{H}_g. \quad (15)$$

**Review-based Interest.** Item reviews serve as a crucial resource for discerning users' genuine intentions and preferences. Therefore, we construct the review-based interest by leveraging the valuable insights conveyed through item reviews. To achieve this, we employ the Transformer to encode the retrieved reviews and learn

their representations. Given a review $\mathcal{R}$, let $\mathcal{H}^{l-1}(\mathcal{R})$ represent the output of embeddings from the previous transformer layer, and $\mathcal{H}^l(\mathcal{R})$ denote the output of the current layer. By leveraging the MHA$(\cdot, \cdot, \cdot)$ function, the review-based interest $\boldsymbol{P}_r$ can be described:

$$\mathcal{H}^{(l)}(\mathcal{R}) = \text{MHA}(\mathcal{H}^{(l-1)}(\mathcal{R}), \mathcal{H}^{(l-1)}(\mathcal{R}), \mathcal{H}^{(l-1)}(\mathcal{R})). \quad (16)$$

$$\boldsymbol{H}_r = \text{MHA}(\mathcal{H}^{(\mathcal{L}_r-1)}(C), \mathcal{H}^{(\mathcal{L}_r-1)}(C), \mathcal{H}^{(\mathcal{L}_r-1)}(C)). \quad (17)$$

$$\boldsymbol{S}_r = \text{Softmax}(\text{S}_{\text{op}}(\frac{\text{SigMo}[(\text{SigMo}(\mathbf{W}_q^{(r)}\boldsymbol{H}_r))(\text{SigMo}(\mathbf{W}_k^{(r)}\boldsymbol{H}_r))^T]}{\sqrt{d}})). \quad (18)$$

$$\boldsymbol{P}_r = \boldsymbol{S}_r \otimes \boldsymbol{H}_r. \quad (19)$$

Here $\mathcal{L}_r$ is the number of transformer layers.

## 3.2 Interest-Enhanced CRS

In this section, we integrate these multi-aspect user interests, namely the context-based interest $\boldsymbol{P}_c$, graph-based interest $\boldsymbol{P}_g$, and review-based interest $\boldsymbol{P}_r$, into the Interest-Enhanced CRS. By doing so, we are able to accurately predict items in the recommendation task and generate dialogue responses effectively in the conversation task.

*3.2.1* **Item Recommender**. In order to enhance the diversity of recommendation results, we leverage these multi-faceted user interests to delve into users' authentic preferences. To achieve this, we concatenate the different user interests, creating the ultimate recommendation-based user references denoted as $\boldsymbol{P}_{\text{rec}}$. Next, we facilitate interactions between $\boldsymbol{P}_{\text{rec}}$ and the feature embeddings of the candidate movie set to compute the rating scores. This approach enables us to effectively predict users' preferences and provide well-informed item recommendations. Formally, this process can be outlined as follows:

$$\boldsymbol{P}_{\text{con}} = [\boldsymbol{P}_c \oplus \boldsymbol{P}_g \oplus \boldsymbol{P}_r];$$
$$\boldsymbol{P}_{\text{rec}} = \text{Softmax}(\text{MLP}(\boldsymbol{P}_{\text{con}})); \quad (20)$$
$$V_{\text{sco}} = \boldsymbol{P}_{\text{rec}} \otimes \boldsymbol{v}.$$

Here $\oplus$ means the concatenation operation, MLP is the Multilayer Perceptron Layer, $\boldsymbol{v}$ represents the feature embedding of the movie item $v$, while $V_{\text{sco}}$ corresponds to the user's rating score for the item $v$. Then, we adopt the cross-entropy to train the recommendation parameters. Formally, the cross-entropy loss $\mathcal{L}_r$ between the prediction $\boldsymbol{P}_{\text{rec}}$ and the target item category can be computed as:

$$\mathcal{L}_r = -\frac{1}{N}\sum_{j=1}^{N} \log P_{\text{rec}}^j, \quad (21)$$

here $N$ is the number of total recommendations and $P_{\text{rec}}^j$ denotes the target category in the $j$-th recommendation.

*3.2.2* **Response Generator**. To effectively generate diverse responses, we incorporate the multi-aspect user interests $\boldsymbol{P}_c$, $\boldsymbol{P}_g$, and $\boldsymbol{P}_r$ into a multi-head attention network to predict the next utterances. The main reason for adopting these attention layers is to seamlessly integrate the entities from the knowledge graph (KG) and reviews into the context information, following the approach of previous work [21]. Furthermore, we augment the attention

mechanism to enhance data representations and filter out noise by leveraging this multi-aspect knowledge, as illustrated below:

$$\mathbf{A}_0^i = \text{MHA}(\mathbf{Y}^{i-1}, \mathbf{Y}^{i-1}, \mathbf{Y}^{i-1}),$$
$$\mathbf{A}_1^i = \text{MHA}(\mathbf{A}_0^i, \mathbf{P}_c, \mathbf{P}_c),$$
$$\mathbf{A}_2^i = \text{MHA}(\mathbf{A}_1^i, \mathbf{P}_g, \mathbf{P}_g), \qquad (22)$$
$$\mathbf{A}_3^i = \text{MHA}(\mathbf{A}_2^i, \mathbf{P}_r, \mathbf{P}_r),$$
$$\mathbf{Y}^i = \text{FFN}(\mathbf{A}_3^i).$$

Here, $\mathbf{Y}^{i-1}$ denotes the output from the previous time step, $\mathbf{Y}^i$ represents the current output, and $\text{MHA}(\mathbf{Q}, \mathbf{K}, \mathbf{V})$ signifies the multi-head attention module, which can be referred to as Eq. (2). Additionally, $\text{FFN}(\cdot)$ corresponds to the fully-connected feed-forward network comprising two linear layers with a ReLU activation [21]. For the conversation task, we employ the cross-entropy loss [21] as the learning objective for response generation. Formally, the loss can be described as:

$$\mathcal{L}_c = -\frac{1}{M} \sum_{t=1}^{M} \log(\text{Prob}(s_t|s_1, \cdots, s_{t-1})), \qquad (23)$$

where $M$ is the number of turns, $s_t$ denotes the $t$-th sentence in the conversation, and the function $\text{Prob}(\cdot)$ means the generation probability $s_t$ of the next token, which can be expressed as:

$$\text{Prob}(s_t|s_1, \cdots, s_{t-1}) = \text{Prob}_v(s_t|\mathbf{Y}_i)$$
$$+ \text{Prob}_g(s_t|\mathbf{Y}_i, \mathcal{G}) \qquad (24)$$
$$+ \text{Prob}_r(s_t|\mathbf{Y}_i, \mathcal{R}),$$

where $\text{Prob}_v(\cdot)$, $\text{Prob}_g(\cdot)$, and $\text{Prob}_r(\cdot)$ are the probability functions over the vocabulary, entities from the knowledge graph $\mathcal{G}$, and reviews $\mathcal{R}$, respectively, following the previous work [15, 21].

## 4 EXPERIMENTS AND ANALYSES

In this section, we conduct experiments to evaluate the performance of CoMoRec on movie datasets and answer the following questions:

- **RQ1:** How does CoMoRec perform compared to state-of-the-art methods in the conversation task?
- **RQ2:** How does CoMoRec perform compared to state-of-the-art methods in the recommendation task?
- **RQ3:** How does CoMoRec enhance the recommendation diversification in the CRS?
- **RQ4:** How do the context-based interest $\boldsymbol{P}_c$, graph-based interest $\boldsymbol{P}_g$, and review-based interest $\boldsymbol{P}_r$ contribute to the performance?

### 4.1 Experimental Protocol

*4.1.1 Datasets.* We evaluate CoMoRec on two widely-adopted movie datasets: **REDIAL** [17] and **TG-REDIAL** [45]. REDIAL is an English dataset for real-world dialogues on movie recommendations, featuring 10,006 conversations about 51,699 movies. It also includes a review database with 30 reviews per movie from IMDb[1] on the previous work [21]. TG-REDIAL is a Chinese conversational recommendation dataset with 10,000 dialogues and 129,392 utterances about 33,834 movies. Each conversation starts with the first

[1]https://www.dbpedia.org/

sentence and progresses to generate responses or recommendations. The review data for TG-REDIAL is sourced from Douban[2].

*4.1.2 Baselines.* To fully evaluate our CoMoRec on these two datasets, we compare our CoMoRec with a series of state-of-the-art methods in both recommendation task and recommendation task. The compared methods include **Trans** [30], **Redial** [17], **KBRD** [7], **KGFS** [43], **KECRS** [40], **RevCore** [21], **KGCR** [24], **$C^2$-CRS** [46], and **MHIM** [26]. By conducting a thorough comparison with these baselines, we can effectively evaluate the performance and effectiveness of our CoMoRec in both tasks.

*4.1.3 Evaluation Metrics.* Our method is comprised of the recommendation and conversation tasks. For the recommendation task, we adopt Recall@$k$ [21, 46] (R@$k$, $k$=1, 10, 50) as the evaluation metrics. For the conversation task, we use automatic evaluation and human evaluation to evaluate the performance of the response generation. For automatic evaluation, we adopt Distinct $n$-gram (D-$n$, $n$=2,3,4) [21, 46] and Bleu-m (B-$m$, $m$=2,3) [34] to evaluate the diversity of generated response contexts at sentence level. Besides, we provide annotators to manually estimate the generated candidates in *Fluency* and *Informativeness*.

## 5 PERFORMANCE COMPARISON

### 5.1 Evaluation on Conversation Task (RQ1)

*5.1.1 Automatic Evaluation.* Table 1 showcases the experimental results, underscoring the superior performance of our model in comparison to other competitive methods. Specifically, in the REDIAL and TG-REDIAL datasets, ReDial surpasses Trans on D-2 by 22.4% and 3.8%, respectively. This success can be attributed to ReDial's utilization of a pre-trained RNN model to enhance representations of past conversations. However, it's noteworthy that both KBRD and KGFS exhibit even greater performance than ReDial on D-2, with a relative improvement of 4.9% and 39.0% in the REDIAL dataset, respectively. The integration of additional information, such as DBpedia, in KBRD and KGFS enhances feature representation learning, leading to enhanced performance. Moreover, RevCore outperforms KBRD by 7.0% on D-2 in the REDIAL dataset. This advancement is a result of RevCore's capability to retrieve pertinent reviews and integrate them into the dialogue context, thereby enriching the overall comprehension of the conversation. Furthermore, $C^2$-CRS surpasses various competitive baselines, including KBRD, KGFS, and RevCore, with improvements of 89.5%, 43.0%, and 77.2% on D-2 in the REDIAL dataset, respectively. The exceptional performance of $C^2$-CRS can be attributed to its innovative contrastive learning-based coarse-to-fine strategy, which effectively merges diverse data representations and enhances dialogue understanding.

From Table 1, it is apparent that our CoMoRec outshines all competing methods on both datasets. For example, on the REDIAL dataset, CoMoRec surpasses Trans, ReDial, KBRD, KGFS, KECRS, RevCore, and $C^2$-CRS by 149.3%, 103.7%, 94.2%, 46.5%, 317.5%, 81.5%, and 2.5% in D-2, respectively. On the TG-REDIAL dataset, CoMoRec outperforms these methods by 262.3%, 249.0%, 326.7%, 123.3%, 308.5%, 346.5%, and 1.6% in D-2, respectively. Notably, CoMoRec also consistently exceeds all state-of-the-art methods in

[2]https://movie.douban.com/

| Datasets | REDIAL | | | | | | | TG-REDIAL | | | | | | |
|---|---|---|---|---|---|---|---|---|---|---|---|---|---|---|
| Models | D-2 | D-3 | D-4 | B-2 | B-3 | Flu. | Inf. | D-2 | D-3 | D-4 | B-2 | B-3 | Flu. | Inf. |
| Trans | 0.067 | 0.139 | 0.227 | 0.0164 | 0.0027 | 0.97 | 0.92 | 0.053 | 0.121 | 0.204 | 0.0335 | 0.0075 | 0.81 | 0.83 |
| ReDial | 0.082 | 0.143 | 0.245 | 0.0198 | 0.0054 | 1.35 | 1.04 | 0.055 | 0.123 | 0.215 | 0.0387 | 0.0094 | 0.98 | 0.101 |
| KBRD | 0.086 | 0.153 | 0.265 | 0.0203 | 0.0061 | 1.23 | 1.15 | 0.045 | 0.096 | 0.233 | 0.0411 | 0.0107 | 0.112 | 0.115 |
| KGSF | 0.114 | 0.204 | 0.282 | 0.0211 | 0.0067 | 1.48 | 1.37 | 0.086 | 0.186 | 0.297 | 0.0442 | 0.0128 | 1.21 | 1.30 |
| KECRS | 0.040 | 0.090 | 0.149 | 0.0124 | 0.0042 | 1.39 | 1.19 | 0.047 | 0.114 | 0.193 | 0.0319 | 0.0053 | 0.86 | 0.90 |
| RevCore | 0.092 | 0.163 | 0.221 | 0.0219 | 0.0083 | 1.52 | 1.34 | 0.043 | 0.105 | 0.175 | 0.0431 | 0.0118 | 1.21 | 1.28 |
| $C^2$-CRS | 0.163 | 0.291 | 0.417 | 0.0223 | 0.0088 | 1.55 | 1.47 | 0.189 | 0.334 | 0.424 | 0.0434 | 0.0120 | 1.25 | 1.37 |
| MHIM | 0.164 | 0.293 | 0.415 | 0.0226 | 0.0089 | **1.60** | 1.44 | 0.186 | 0.333 | 0.426 | 0.0435 | 0.0118 | 1.28 | 1.35 |
| **CoMoRec** | **0.167*** | **0.298*** | **0.421*** | **0.0230*** | **0.0089*** | 1.57 | **1.51*** | **0.192*** | **0.342*** | **0.428*** | **0.0437*** | **0.0125*** | **1.32*** | **1.39*** |

**Table 1: Results on the conversation task. Flu. and Inf. stand for Fluency and Informativeness, respectively. Numbers marked with * denote that there is a statistically significant improvement compared with the best baseline (t-test with p-value < 0.05).**

both B-2 and B-3 metrics, showcasing the effectiveness of our approach for response generation. The improvement of CoMoRec can be attributed to its focus on enhancing response diversity by modeling multi-aspect user interests by user-system dialogues. By comprehensively considering and capturing various facets of user preferences and interests, CoMoRec enhances its capacity to generate diverse and personalized responses, thereby achieving superior performance compared to other baselines.

*5.1.2 Human Evaluation.* Table 1 encapsulates the results of the human evaluation in the conversation task in terms of *Fluency* and *Informativeness* metrics. There are several key observations: 1) Re-Dial demonstrates superior performance over Transformer, owing to the implementation of a pre-trained RNN encoder. This encoder significantly enhances the quality and fluency of the responses generated by the system. 2) KGSF excels in terms of *Informativeness* compared to various other baselines. This achievement can be attributed to its integration of an external information knowledge graph, which effectively aligns the semantics of the conversation context with the items discussed. 3) RevCore achieves the highest level of performance in terms of *Fluency* when compared to several other baselines. This success can be attributed to its utilization of additional reviews to enhance the decoder, resulting in the generation of more coherent and fluent responses. 4) $C^2$-CRS emerges as the top performer in terms of *Informativeness* among all the baselines. The efficacy of $C^2$-CRS can be credited to its emphasis on integrating diverse data types to generate informative words and entities. Notably, CoMoRec consistently outperforms all the compared methods in terms of both metrics. This success can be attributed to its utilization of multi-aspect knowledge to model various levels of user interests, enabling the generation of fluent and diverse responses.

## 5.2 Evaluation on Recommendation Task (RQ2)

Table 2 summarizes the experimental results on the recommendation task for item prediction. The results clearly demonstrate the superiority of our model over all the baselines. Firstly, the results show that both KBRD and KGFS outperform ReDial. For instance, KGFS surpasses ReDial on R@1 by 62.5% and 100.0% on the REDIAL and TG-REDIAL datasets, respectively. This improvement can be attributed to the integration of external information, such as DBpedia, which enriches the representations of items and words in both

| Datasets | REDIAL | | | TG-REDIAL | | |
|---|---|---|---|---|---|---|
| Models | R@1 | R@10 | R@50 | R@1 | R@10 | R@50 |
| ReDial | 0.024 | 0.140 | 0.320 | 0.000 | 0.002 | 0.013 |
| KBRD | 0.031 | 0.150 | 0.336 | 0.005 | 0.032 | 0.077 |
| KGSF | 0.039 | 0.183 | 0.378 | 0.005 | 0.030 | 0.074 |
| KECRS | 0.021 | 0.143 | 0.340 | 0.002 | 0.026 | 0.069 |
| RevCore | 0.046 | 0.220 | 0.396 | 0.004 | 0.029 | 0.075 |
| KGCR | 0.040 | 0.191 | 0.384 | 0.004 | 0.033 | 0.076 |
| $C^2$-CRS | 0.053 | **0.233** | 0.407 | 0.007 | 0.032 | 0.078 |
| MHIM | 0.052 | 0.230 | 0.405 | 0.007 | 0.033 | 0.079 |
| **CoMoRec** | **0.056*** | 0.231 | **0.411*** | **0.009*** | **0.035*** | **0.081*** |

**Table 2: Results on the recommendation task. Numbers marked with * denote that there is a statistically significant improvement compared with the best baseline (t-test with p-value < 0.05).**

KBRD and KGFS. Additionally, both KBRD and KGFS outperform KECRS on R@1 by 47.6% and 85.7% on the REDIAL datasets, respectively. Furthermore, RevCore demonstrates better performance than KBRD and KGFS, achieving a gain of approximately 48.4% and 17.9% on R@1 in the REDIAL dataset, respectively. The primary reason behind this improvement is that RevCore incorporates reviews to enhance the user vector representation. Moreover, $C^2$-CRS outperforms RevCore by 15.2% and 75.0% on R@1 in the REDIAL and TG-REDIAL datasets, respectively. This improvement can be attributed to the utilization of contrastive learning, which facilitates the fusion of multi-type data representations in $C^2$-CRS.

Note that our CoMoRec achieves the best performance among the state-of-the-art methods. Concretely, CoMoRec outperforms RevCore on R@1 by 21.7%, and 125.0% on REDIAL and TG-REDIAL datasets, respectively. CoMoRec is also superior to $C^2$-CRS on R@1 by 5.7%, and 28.6% on REDIAL and TG-REDIAL datasets, respectively. The results demonstrate the effectiveness of our proposed method for item prediction in the recommendation task. This is due to the fact that CoMoRec not only considers the dynamic temporal knowledge graph to address the sparsity of the traditional user-item bipartite graph but also models multi-aspect user interests including context-based interest, graph-based interest, and review-based interest for enhancing the recommendation diversification as users interact with the system over time in the CRS.

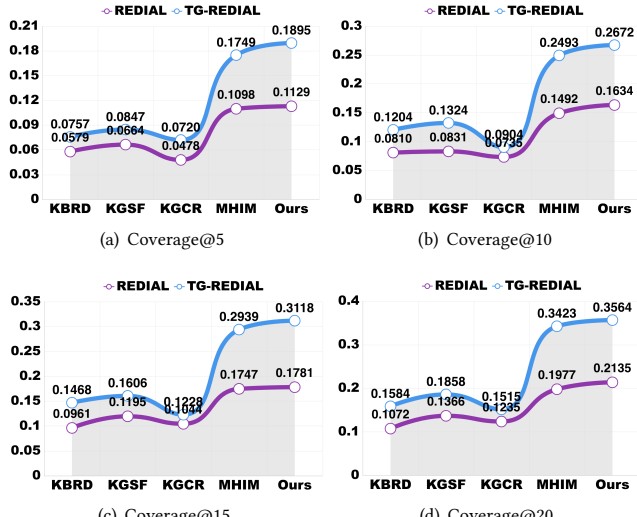

(a) Coverage@5

(b) Coverage@10

(c) Coverage@15

(d) Coverage@20

Figure 2: Results on *Coverage* metrics.

## 5.3 Study on Recommendation Diversity (RQ3)

Given our primary objective of enhancing recommendation diversity as users engage with the system in the CRS, we meticulously analyze the recommendation outcomes and conduct a comprehensive comparison with the strongest baselines to assess the effectiveness of CoMoRec in achieving this goal. Along this line, we utilized the widely recognized metric *Coverage*@k (k=5, 10, 15, 20) to quantify the level of recommendation diversification and account for variations among the recommended items. This well-established metric allowed us to measure the extent to which our recommendations spanned a broad spectrum of the recommendation space. A higher coverage value indicates a greater capacity to encompass items from diverse categories. It signifies the ability of our system to provide recommendations that cover a wide range of item types, catering to varying user preferences and interests.

Figure 2 shows that it consistently achieves the highest *Coverage* values across all datasets when compared to the competitive baselines, validating the superiority of our CoMoRec in diversified recommendation. On the TG-REDIAL dataset, our CoMoRec exhibits significant improvements of 121.93%, 101.81%, 195.58%, and 7.18% in terms of *Coverage@10* when compared to the robust models KBRD, KGSF, KGCR, and MHIM, respectively. These compelling results underscore the effectiveness of CoMoRec in effectively mitigating isolation concerns by ensuring the comprehensive coverage of recommended items. This, in turn, provides users with a broader spectrum of choices, enhancing their overall experience. Consequently, this reinforces CoMoRec's pivotal role in enhancing recommendation diversification in the dynamic user-system feedback loop as users interact with the system over time.

## 5.4 Ablation Studies (RQ4)

In this part, we conduct ablation experiments with different variants of our CoMoRec to verify the contributions of each component designed in our method on the REDIAL dataset, including: 1) CoMoRec

| Models | R@1 | R@10 | R@50 |
|---|---|---|---|
| CoMoRec | **0.056** | **0.231** | **0.411** |
| CoMoRec w/o $P_c$ | 0.052 | 0.225 | 0.408 |
| CoMoRec w/o $P_g$ | 0.054 | 0.228 | 0.406 |
| CoMoRec w/o $P_r$ | 0.055 | 0.230 | 0.404 |

Table 3: Ablation studies on the recommendation task.

| Models | D-2 | D-3 | D-4 |
|---|---|---|---|
| CoMoRec | **0.167** | **0.298** | **0.421** |
| CoMoRec w/o $P_c$ | 0.157 | 0.265 | 0.413 |
| CoMoRec w/o $P_g$ | 0.161 | 0.278 | 0.419 |
| CoMoRec w/o $P_r$ | 0.164 | 0.282 | 0.420 |

Table 4: Ablation studies on the conversation task.

w/o $P_c$: we remove the context-based interest $P_c$; 2) CoMoRec w/o $P_g$: we remove the graph-based interest $P_g$; 3) CoMoRec w/o $P_r$: we remove the review-based interest $P_r$.

Table 3 and 4 shows the experimental results of the ablation studies on both tasks, yielding several significant insights: 1) In the recommendation task, CoMoRec consistently outperforms other models across metrics such as R@1, R@10, and R@50. By effectively integrating the influences of different components ($P_c$, $P_g$, and $P_r$), CoMoRec adeptly captures multi-aspect user interests, resulting in more precise and personalized recommendations. 2) In the conversation task, CoMoRec achieves op scores in evaluation metrics like D-2, D-3, and D-4. Through the synergistic combination of $P_c$, $P_g$, and $P_r$, CoMoRec delivers high-quality and coherent dialog responses, outperforming models that overlook specific components. 3) Ablation studies on CoMoRec underscore the importance of each model component. $P_c$ significantly contributes to performance in both recommendation and conversational tasks, while $P_g$ has a moderate impact on the recommendation task. Additionally, $P_r$ also plays a role in the recommendation task. These insights deepen our understanding of the contributions of individual components within CoMoRec, paving the way for further optimization and enhancement of recommendation and conversational systems.

## 6 CONCLUSION

In the CRS, the lack of diversity will gradually exacerbate as users interact with the system over time, inevitably posing a series of challenges such as filter bubbles and echo chambers. To address these issues, we proposed a novel paradigm, CoMoRec, which consists of User-Centric Multi-Interest Learning and Interest-Enhanced CRS. The former aims to model various facets of user interests, including context-based interest, graph-based interest, and review-based interest, to explore the wide array of user intentions and enrich the diversity of results in conversational movie recommendations; the latter focuses on employing these multiple user interests to accurately predict personalized and diverse movie recommendations in the recommendation task and effectively generate the fluent and informative responses in the conversation task. Extensive experiments on two publicly CRS-based movie datasets show that our CoMoRec achieves a new state-of-the-art performance, and the superior of improving recommendation diversification in the CRS.

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
