# OpenReview forum: "Diversity Matters: User-Centric Multi-Interest Learning for Conversational Movie Recommendation"
_acmmm.org/ACMMM/2024/Conference — MM2024 Poster_

### Official Review · Reviewer_EQk3 · 2024-05-15

**Rating:** 3
**Confidence:** 3

**Summary:**

This paper proposes an innovative conversational recommender system framework called CoMoRec, designed to enhance the diversity of movie recommendations through natural language conversations. CoMoRec consists of two main components: User-Centric Multi-Interest Learning and Interest-Enhanced CRS. The former models user interests by constructing a Temporal Knowledge Graph (TKG) and capturing interests from context, graphs, and reviews, while the latter leverages these interests to make personalized and diverse movie recommendations and generate informative conversational responses. Through experiments on two publicly available movie recommendation datasets, CoMoRec has achieved groundbreaking results in improving recommendation diversity and conversation task performance, demonstrating its effectiveness in enhancing recommendation diversity within the dynamic user-system feedback loop.

**Strengths:**

* **+Clear problem definition**: The article provides detailed explanations for the mathematical representations and formulas used.
* **+Detailed experiments**: The article conducts detailed experiments on the proposed model, selecting many representative baselines for comparison and performing ablation studies.
* **+Good narrative**. The article's motivation and problem background are rational, and it holds significant importance for the research community in the field of recommendations.

**Limitations:**

* **-Insufficient performance comparison metrics**. In terms of recommendation performance, the authors only used recall as the metric. However, common metrics in the recommendation task also include nDCG (normalized Discounted Cumulative Gain) and precision. Did the authors compare the performance of the proposed model with the baseline models using these metrics? I believe that the lack of comparative performance metrics is a significant technical flaw, and therefore, I am inclined to reject this article. However, if the authors can supplement the nDCG and precision scores of the proposed model against the baseline models and demonstrate the superiority of the proposed model, I may accept the article.
* -The use of line charts for the "Results on Coverage metrics" is a bit weird. Perhaps bar charts would be more appropriate.

**Suitability:**

3

---

### Official Review · Reviewer_cpYn · 2024-05-21

**Rating:** 4
**Confidence:** 3

**Summary:**

The authors propose CoMoRec, a novel framework for conversational recommendation systems. CoMoRec considers a dynamic temporal knowledge graph and models multi-aspect user interests, including context-based, graph-based, and review-based interests for enhancing recommendation diversification.

**Strengths:**

1.The paper is well-written and easy to read.

2.The framework considers multi-aspect user interests.

3.Extensive experiments are provided.

**Limitations:**

1.Some typos.  There are some typos present in the paper, such as a missing parenthesis in equation (2) and a duplicate phrase in line 643.

2.Performance improvements.  The improvements over the strongest baselines, MHIM and $C^2-CRS$, are not notably significant, which raises the question of whether a strong baseline, such as MHIM, augmented with review-based information could outperform the proposed CoMoRec.

3.Comparison with LLMs. Given the recent success of large language models (LLMs) in developing powerful conversational recommender systems (CRSs), it would be better for the authors to discuss the advantages of the proposed framework over LLMs in the context of conversational recommender systems.

4.Metric explanation. The metrics used to evaluate the system, Fluency and Informativeness, are not detailed or explained sufficiently. It would be better for the authors to provide a clearer definition and rationale for these metrics, as well as explain how they relate to the overall performance of the system.

**Suitability:**

3

---

### Official Review · Reviewer_LneQ · 2024-05-25

**Rating:** 2
**Confidence:** 3

**Summary:**

The paper addresses the critical issue of diversity in Conversational Recommender Systems (CRSs), emphasizing that current approaches often result in repetitive content, leading to phenomena like filter bubbles and echo chambers. To tackle this, the authors propose a novel end-to-end system called User-Centric Multi-Interest Learning for Conversational Movie Recommendation (CoMoRec). CoMoRec aims to enhance diversity by automatically modeling user interests from multiple perspectives—context-based, graph-based, and review-based—during natural language conversations. By leveraging these multi-aspect interests, the system can predict personalized and diverse movie recommendations and generate informative responses.

**Strengths:**

The problem addressed by CoMoRec is both important and highly relevant in today's digital landscape.
Extensive experiments conducted on two publicly available CRS-based movie datasets demonstrate that CoMoRec achieves state-of-the-art performance. It outperforms existing methods in both the recommendation and conversation tasks, providing more accurate, personalized, and diverse movie recommendations while generating fluent and informative conversational responses.

**Limitations:**

- The paper's writing suffers from significant repetition, which detracts from its clarity and conciseness. For instance, the authors repeatedly emphasize the importance of diversity in recommender systems and the issues of filter bubbles and echo chambers, reiterating these points multiple times throughout the abstract and introduction.
- The notation used in the paper is overly complex and extensive, which significantly impacts its readability. The excessive use of symbols, multiple layers of subscripted and superscripted variables, and intricate mathematical expressions make it challenging for readers to follow the core ideas and methodologies. This level of detail, while aiming for precision, can obscure the main contributions and makes comprehension difficult. Simplifying the notation and providing summary tables or diagrams could greatly enhance clarity, making the paper more accessible to a broader audience, including those less familiar with the specific technical nuances.
- While CoMoRec presents an innovative approach to enhancing recommendation diversity in CRS, it is important to note that the method's complexity and reliance on existing building blocks, such as Temporal Knowledge Graphs and Relational Graph Convolutional Networks, render it more like an engineering feat rather than groundbreaking research.
- Below are a few papers that I think should be discussed in the related work:
    - Shengnan Lyu, Arpit Rana, Scott Sanner, Mohamed Reda Bouadjenek: A Workflow Analysis of Context-driven Conversational Recommendation. WWW 2021: 866-877
    - Tianshu Shen, Jiaru Li, Mohamed Reda Bouadjenek, Zheda Mai, Scott Sanner: Towards understanding and mitigating unintended biases in language model-driven conversational recommendation. Inf. Process. Manag. 60(1): 103139 (2023)

**Suitability:**

1

---

### Meta-Review · Area_Chair_2z4o · 2024-07-05

**Recommendation:** Accept (Poster)
**Confidence:** 4

**Metareview:**

The paper proposes a novel framework, CoMoRec, for a conversational movie recommendation, enhancing diversity by modeling multi-aspect user interests. Reviewers praise the paper's importance, clarity, and experimental rigor but raise concerns about notation complexity, comparison with large language models, and performance metrics. The authors' rebuttal addresses these concerns, providing additional explanations and results. While some reviewers suggest minor improvements, the overall assessment indicates a valuable contribution to the multimedia community, warranting presentation as a poster.